# Residential green space and child intelligence and behavior across urban, suburban, and rural areas in Belgium: A longitudinal birth cohort study of twins

Esmée M. Bijnens[1,2]*, Catherine Derom[2,3], Evert Thiery[4], Steven Weyers[2], Tim S. Nawrot[1,5]

1 Centre for Environmental Sciences, Hasselt University, Diepenbeek, Belgium, 2 Department of Obstetrics and Gynaecology, Ghent University Hospital, Ghent, Belgium, 3 Centre of Human Genetics, University Hospitals Leuven, Leuven, Belgium, 4 Department of Neurology, Ghent University Hospital, Ghent, Belgium, 5 Department of Public Health, Leuven University (KU Leuven), Leuven, Belgium

* esmee.bijnens@uhasselt.be

**Data Availability Statement:** Green space data (CORINE land cover) used in this study can be

## Abstract

### Background

Exposure to green space has beneficial effects on several cognitive and behavioral aspects. However, to our knowledge, no study addressed intelligence as outcome. We investigated whether the level of urbanicity can modify the association of residential green space with intelligence and behavior in children.

### Methods and findings

This study includes 620 children and is part of the East Flanders Prospective Twin Survey (EFPTS), a registry of multiple births in the province of East Flanders, Belgium. Intelligence was assessed with the Wechsler Intelligence Scale for Children-Revised (WISC-R) in 620 children (310 twin pairs) between 7 and 15 years old. From a subset of 442 children, behavior was determined based on the Achenbach Child Behavior Checklist (CBCL). Prenatal and childhood residential addresses were geocoded and used to assign green space indicators. Mixed modeling was performed to investigate green space in association with intelligence and behavior while adjusting for potential confounding factors including sex, age, parental education, neighborhood household income, year of assessment, and zygosity and chorionicity.

We found that residential green space in association with both intelligence and behavior in children was modified by the degree of urbanicity (*p* < 0.001). In children living in an urban environment, multivariable adjusted mixed modeling analysis revealed that an IQR increment of residential green space (3,000-m radius) was associated with a 2.6 points (95% CI 1.4–3.9; *p* < 0.001) higher total intelligence quotient (IQ) and 2.0 points (95% CI −3.5 to −0.4; *p* = 0.017) lower externalizing behavioral score. In children residing in a rural or suburban environment, no association was found. A limitation of this study is that no information

obtained from the European Environment Agency (EEA) (https://land.copernicus.eu/pan-european/corine-land-cover). Air pollution data used in this study can be obtained from IRCEL (https://www.irceline.be/en). Access to and use of summarized health data of study participants requires submission of a research proposal to the East Flanders Prospective Twin Survey, Ghent University, contact email, twins@uzgent.be.

**Funding:** Dr. Bijnens holds a fellowship from the Marguerite-Marie Delacroix foundation. Since its start, the East Flanders Prospective Twin Survey has been partly supported by grants from the Fund of Scientific Research Flanders and Twins, a nonprofit Association for Scientific Research in Multiple Births (Belgium). The funders had no role in study design, data collection and analysis, decision to publish, or preparation of the manuscript.

**Competing interests:** The authors have declared that no competing interests exist.

**Abbreviations:** CBCL, Child Behavior Checklist; CORINE, Coordination of Information of the Environment; IQ, intelligence quotient; IQR, interquartile range; NDVI, normalized difference vegetation index; PAH, airborne polycyclic aromatic hydrocarbon; PIQ, performance intelligence quotient; SDQ, Strengths and Difficulties Questionnaire; TIQ, total intelligence quotient; VIQ, verbal intelligence quotient; WISC-R, Wechsler Intelligence Scale for Children-Revised.

was available on school location and the potential for unmeasured confounding (e.g., time spend outdoors).

## Conclusions

Our results indicate that residential green space may be beneficial for the intellectual and the behavioral development of children living in urban areas. These findings are relevant for policy makers and urban planners to create an optimal environment for children to develop their full potential.

## Author summary

### Why was this study done?

- This study examines residential surrounding green space in association with intelligence and behavior in a study area that includes a spectrum of urban to rural environments.
- Previous studies mainly focused on urban areas, whereas only a few studies have explored differences in the effect of green space on cognition in nonurban settings.
- Understanding the health disparities that exist between urban and rural environments is essential for maintaining and improving human well-being in a rapidly urbanizing world.

### What did the researchers do and find?

- This longitudinal birth cohort study of twins assessed intelligence in 620 children between 7 and 15 years old.
- Our results indicate that residential green space is especially beneficial for intellectual and behavioral development of children living in an urban environment.

### What do these findings mean?

- We show that low residential green space in urban children is associated with a "shift" towards a higher incidence of low IQ demonstrating the public health impact of our findings.

## Introduction

The brain develops steadily during prenatal and early postnatal periods, which are considered as the most vulnerable windows for effects of environmental exposures [1]. Prenatal exposure to air pollutants may have persisting effects on brain development in children with consequences on cognition and behavior [2]. Later, across childhood and adolescence, crucial changes in cognition, emotion, and behavior occur [3], and some specific functions related

with learning and school achievement develop [4]. Environmental factors such as air pollution are associated with neuroinflammation and may impact these performances in healthy children [3]. Children residing in a highly polluted urban environment exhibited deficits in fluid intelligence, memory, executive functions, and also impairments in subscales of intelligence quotient (IQ), relative to children living in a less polluted urban environment [3]. Moreover, airborne polycyclic aromatic hydrocarbons (PAHs) are generated by the incomplete combustion of fossil fuels, and prenatal exposure is significantly associated with a reduced IQ at 5 years of age [5–7]. However, the effects of green space on intelligence are currently unknown.

Greenness refers to the vegetation level, whereas green spaces are covered partly by grass, trees, or other vegetation, and include city parks, community gardens, sports fields, as well as natural and forested areas in rural environments [8,9]. Green spaces provide environmental benefits by reducing air and noise pollution. Higher residential green space is associated with lower ambient air pollution [10] and noise exposure [11]. Both have been associated with diminished cognitive development [12–15]. Furthermore, from a health perspective, green space promotes physical activity [16,17] and stress reduction [18]. Apart from the promotion of physical activity, green space in the living environment could also lead to more social contacts [19]. In addition, higher levels of neighborhood green space were associated with significantly lower levels of symptoms of depression, anxiety, and stress.[20] Besides self-reported stress, also salivary cortisol, an indicator of chronic stress, is negatively associated with exposure to green space in the living environment [21]. Higher residential green space during pregnancy is even associated with longer telomere length in placental tissue, an important marker of ageing [22]. Finally, according to the biodiversity hypothesis of health, the reduced contact with a natural environment and biodiversity is associated with adverse implications for the human microbiome and its immunomodulating capacity [23]. Microbial input from the environment may drive brain regulation and is a possible major component of the beneficial effect of greenness [24]. Several animal studies suggest that the human microbiome may have profound effects on brain development and linked the gut microbiota to neurodevelopmental processes and behavior [25–27]. Natural space-induced environmental biodiversity has been related to enriched skin and gut microbiota that, in turn, has been shown to improve brain development, probably through regulation of immune response [28].

In a previous study of 2,593 children attending primary school in Barcelona [29], exposure to surrounding green space at the home and school area was associated with greater progress in working memory and attention [29]. The literature on green space around schools and academic performance shows mixed results [30]. Although positive associations were observed for 3 types of green space: green land cover (45% of the findings), greenness (32%), and tree cover (30%), whereas no positive findings were found for agriculture, grass, or shrub cover. Furthermore, access to nearby natural outdoor play spaces may enhance children's executive functioning [31], and a meta-analysis revealed that the greening of schoolyards across multiple sites in North America and Western Europe has been associated with better psychological well-being and improved school performance among pupils [32].

Evidence shows that green not only results in a beneficial effect on cognitive and school performances but also in improved psychological well-being with positive effects on emotions and behavior [33–35]. Higher residential surrounding green space was significant inversely associated with behavioral development in 2,111 schoolchildren from 36 schools in Barcelona [36]. Furthermore, residential green space was associated with reduced scores for hyperactivity and inattention symptoms [36]. A study in 1,932 children residing in the city of Munich and its surrounding areas showed that a higher distance between a child's residence and the nearest urban green space was positively associated with hyperactivity/inattention (odds ratio = 1.20 per 500-m increase in distance) [37]. In addition, reduced aggressive behavior was noted in

association with more surrounding residential green space in urban-dwelling adolescents [38]. Few studies on green space and cognition [39] or behavior [37,40] have investigated effect modification by urbanicity but did not find statistically significant differences by urbanization of residence.

The research mentioned previously investigated green space in association with cognitive and behavioral aspects; in connection to this, we investigated the association between green space and intelligence. A major limitation is that most studies mainly focused on urban areas, whereas only a few studies have explored differences in the effect of green space on cognition in nonurban settings. Understanding the disparities that exist between urban and rural environments is important for building and sustaining an effective policy. This study examines the relation of environmental green space to intelligence and behavior in a study area that includes a spectrum of urban to rural environments across the province of East Flanders in Belgium. We hypothesize that the level of urbanicity can modify the outcome on intelligence and behavior in children associated with green space exposure in early life.

## Methods

### Subject recruitment

This study includes 620 children born between 1980 and 1991. This subgroup is part of the East Flanders Prospective Twin Survey (EFPTS), a registry of multiple births in the province of East Flanders, Belgium. Since its start in 1964, over 10,000 twin pairs have been registered [41,42]. The province East Flanders is located in the North-West of Belgium, and the capital Ghent is centrally located. Besides Ghent and a few urban areas, most of the province is rural (S1 Fig). Similar to the rest of Flanders, this province is densely populated with strong urban metropolitan areas and includes a lot of arable land. The capital Ghent has many parks spread throughout the city, readily accessible or within close proximity to home for most residents [43]. Parks in Flanders are used mostly as a place to relax rather than to engage in intensive sports [43,44].

In 1992 and between 1996 and 1999, we assessed intelligence in 663 twin pairs at a mean age of 10.4 years (age range 7–15) using the Wechsler Intelligence Scale for Children-Revised (WISC-R). Of the 663 pairs, we invited 599 eligible mothers to complete a questionnaire (S1 Text) on residential history. In total, 64 mother/twin pairs were lost for the present study: mothers with missing contact details ($n$ = 7), mothers who indicated not willing to further participate ($n$ = 20), late mothers ($n$ = 19), and twin pairs of which one of the twins died (still born or neonatal, $n$ = 18). The questionnaire was completed by 324 mothers (54%), and 275 mothers must be classified as nonresponders, i.e., 126 could not be reached, 126 did not fill out the questionnaire, and 23 indicated not willing to participate. We excluded 14 twin pairs from our analysis because of missing data (on residential location [$n$ = 5] and parental educational level [$n$ = 9]). The number of children included in our analysis was 620 individuals (310 pairs). From a subset of 442 individuals (221 twin pairs), aged 6–15 years, data are available on behavioral analysis. An overview is presented in the flowchart as supplement (S2 Fig). We compared the characteristics of those that we were able to collect information on residential history compared with nonresponders. Children included in the study differed slightly from those of which we were not able to collect data in terms of parental educational level, IQ, and externalizing behavior. However, given that the differences between responders and nonresponders were rather small, it is unlikely that this would have had a major impact on the results. This study is reported as per the Strengthening the Reporting of Observational Studies in Epidemiology (STROBE) guideline (S1 STROBE Checklist). The method and study design used in this study were described before the analyses were conducted, and the prospective analysis plan is

present in supplement (S2 Text). Written informed consent was obtained from all participants, and ethical approval was given by the Ethics Committee of University Hospital Ghent and Hasselt University (Registration number: B670201730788).

## Data collection

Data recorded by the obstetrician at birth included gestational age, birth weight, and sex of the twins. Gestational age was calculated as the number of completed weeks of pregnancy and was estimated on the basis of last menstrual period combined with real-time ultrasonography in early pregnancy. At time of birth, placentas were examined within 24 hours after delivery by a trained midwife following a standardized protocol [45]. Cord insertion categorized into 2 groups: central insertion (central, paracentral, paramarginal) and peripheral insertion (marginal, membrane septum, and membrane peripheral). Zygosity was determined by sequential analysis based on sex, choriontype, blood group determined on umbilical cord blood, placental alkaline phosphatase, and since 1982, DNA fingerprints [46]. After DNA fingerprinting, a zygosity probability of 99.9% was reached. Based on chorionicity, monozygotic twins where further divided into 2 groups, monozygotic dichorionic and monozygotic monochorionic. Monochorionic twins share only 1 placenta [47].

Further, we collected information on parental education. The highest educational level of both parents as a proxy of socioeconomic status (SES) was categorized into 2 groups according to the Belgian education system: (1) no education, primary school, lower secondary education, or higher secondary education, and (2) tertiary education. We gathered information on neighborhood socioeconomic status. Based on the home addresses, all twins were assigned to statistical sectors (average area = 1.55 km$^2$), the smallest administrative entity for which statistical data are produced by the Belgian National Institute of Statistics (NIS). Belgian census data (FOD Economie/DG Statistiek) derived from the NIS were used to define neighborhood household income based on annual household income in the year 1994.

## Green space and traffic-related exposure

Residential addresses of the children during pregnancy and during childhood at time of the IQ measurement were geocoded. All analyses were carried out using Geographic Information System (GIS) functions with ArcGIS 10 software. Residential location was categorized into urban, suburban, and rural based on a map (Flemish Government–Department Environment) containing all statistical sectors in Flanders classified as urban, suburban, and rural areas depending on their population density, employment, location, and spatial planning (S1 Fig). Seminatural, forested, blue, and urban green areas (green space) in several radius distances (5,000, 4,000, 3,000, 2,000, 1,000, and 500 m) around the residential address during childhood and during pregnancy were estimated based on CORINE (Coordination of Information of the Environment) Land Cover 1990 (European Environment Agency). Green space does not include agricultural areas. Moreover, we calculated residential green space based on a high-resolution (1 m$^2$) land cover data set (Green Map of Flanders 2012 from the Agency for Geographic Information Flanders [AGIV]), which is available. We estimated high green (vegetation height higher than 3 m), within several radii (2,000, 1,000, 500, 300, 100, and 50 m) around the residence.

Residential distances to the nearest major road were determined. Major roads were defined by 2 types of roads, namely, freeways and national roads. We calculated the regional background levels of $NO_2$ for the residential address during childhood using a kriging interpolation method [48,49] that uses land cover data obtained from satellite images. This model provides interpolated $NO_2$ values from the Belgian telemetric air quality networks in 4 × 4 km grids. We calculated individual $NO_2$ concentrations (micrograms per cubic meter) 1 year before the measurement.

## Assessment of intelligence and behavior during childhood

Intelligence was assessed by the WISC-R. This test was administered separately and at random by 2 trained research workers [50]. The methods of assessment of intelligence has been previously described [51]. In brief, the WISC-R consists of 6 verbal and 6 performance subscales and has been validated for use in this population [52]. The scores on the subscales are standardized for age and added up to verbal IQ (VIQ), performance IQ (PIQ), and total IQ (TIQ). The minimum and maximum scores that can be obtained are 50 and 150 points. In this study, the VIQ, PIQ, and TIQ scores were analyzed.

Behavior was assessed by the Achenbach Child Behavior Checklist (CBCL). This checklist was developed by Achenbach (1991) to examine the extent to which children have behavioral and emotional problems as perceived by their parents [53]. Good reliability and validity have been demonstrated for the Dutch CBCL [54]. Parents were asked to answer each item in 3 levels: 0 = not true, 1 = sometimes/somewhat true, and 2 = very true/often true, with higher score indicating more adverse behavior. The sum of all questions served as a global score for children's problematic behavior (the total problems behavior score). Separate scores were calculated for internalizing problems and externalizing problems. CBCL T-scores equal to 64 or higher on the internalizing, externalizing, or total problem scales are considered in the clinical range [55]. In this study, we examined the total behavioral score, and the externalizing and internalizing subdomains. The internalizing behavioral score is formed by combining emotionally reactive, anxious/depressed, somatic complaints, and withdrawn. The externalizing behavioral score is formed by combining attention problems and aggressive behavior.

## Statistical analysis

For data management and statistical analyses, we used SAS software, version 9.4 (SAS Institute, Cary, NC). All reported $p$-values are 2-sided and were considered statistically significant when $p < 0.05$. The distribution of all variables was inspected. Mixed modeling was performed to investigate intelligence/behavior in association with residential green space. The twins were analyzed as individuals in a multilevel regression analysis to account for relatedness between twin members by adding a random intercept to the model. The variance–covariance structure was allowed to differ between the 3 zygosity and chorionicity groups, including dizygotic dichorionic, monozygotic dichorionic, and monozygotic monochorionic. Mixed modeling was performed adjusted for covariates selected a priori. The models investigating the association between intelligence/behavior and green space during childhood included the covariates sex, age, parental education, neighborhood household income, year of assessment WISC-R/CBCL, and zygosity and chorionicity. We expressed the estimates for an IQR change in percentage green space for buffers from 1,000 to 5,000 m. Because of the skewedness of the data for the 500-m radius, we expressed the effect size for a dichotomized variable representing the presence or absence of green space within this buffer. To capture potential nonlinear effects of year of assessment, the quadratic terms of this variable was tested but did not reach significance. Further, to allow for nonlinearity, green space variables were categorized into tertiles. We constructed scatterplots (Fig 1), and in addition, we plotted IQ (TIQ, VIQ, and PIQ) for each tertile of green space (3,000-m buffer). We did not find evidence for deviation from linearity. Studying green space during pregnancy additional adjustment were made for perinatal factors such as birth weight, gestational age, birth year (linear and quadratic), cord insertion, and maternal smoking during pregnancy. We tested the interaction of urbanicity-by-green space interaction on IQ. Because of a significant interaction, we stratified the analysis in children residing in an urban, suburban, and rural area at time of the IQ measurements. To address the public health

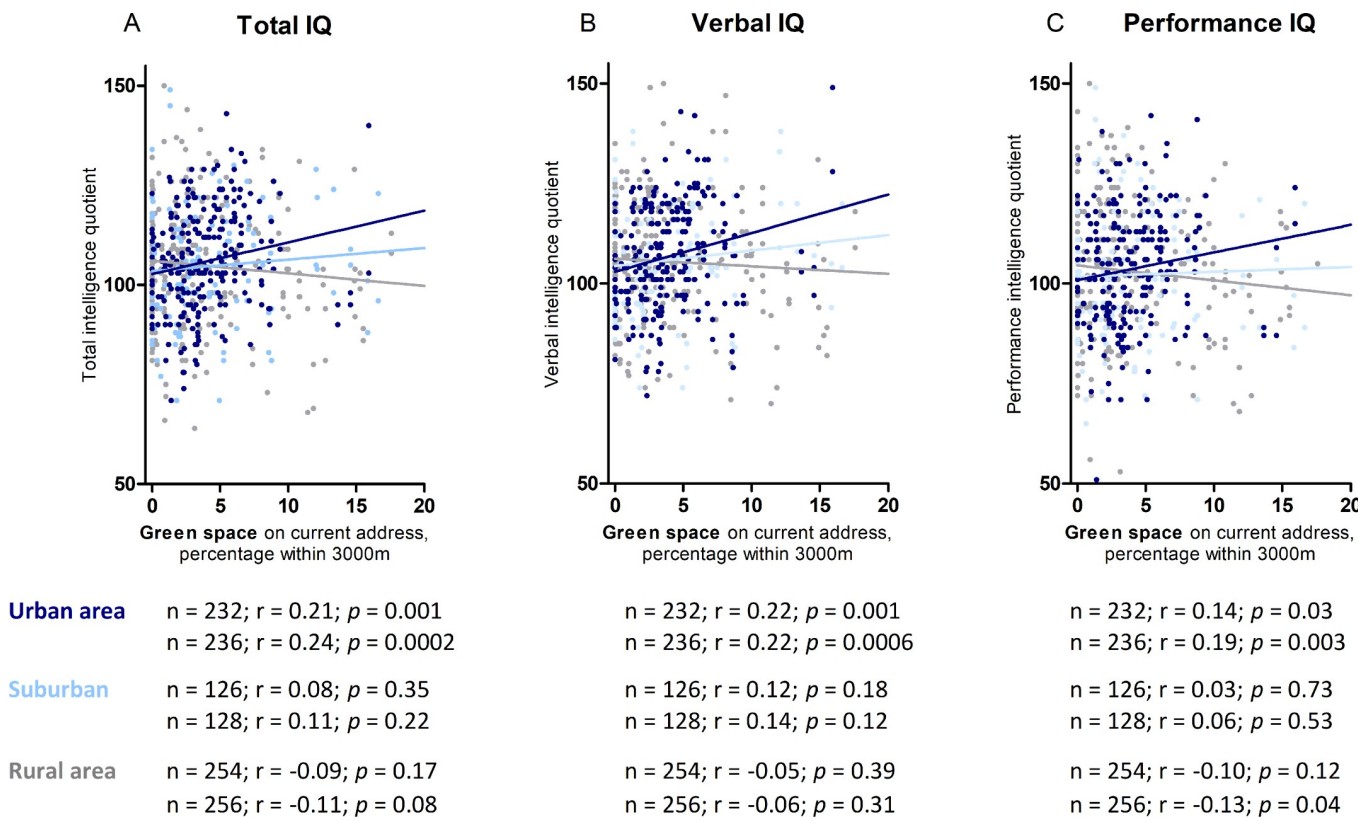

**Fig 1. The association between intelligence and residential green space is modified by the degree of urbanicity (*p* for interaction < 0.001).** Intelligence is shown in association with green space in a 3,000-m radius around the current residence in twins living in an urban (*n* = 232), suburban (*n* = 126), and a rural area (*n* = 254). Data points (*n* = 8) with green space >20% were excluded from the figure for visibility of the figure. (A) Total IQ, (B) verbal IQ, and (C) performance IQ. Correlation coefficients are shown for data points in figure (excluding points with green space >20%) and for all data points. IQ, intelligence quotient.

significance, we compared the distribution of total IQ for children residing in "high" (above median) versus "low" (below median) residential green space across the 3 urbanization levels.

Finally, we conducted a series of sensitivity analysis to check the robustness of the findings. First, as a sensitivity analysis, we investigated the association between green space and IQ in a subset of children (*n* = 144) living minimum 10 years (or their whole life if the age of the child <10 years old) at the current address. Second, we adjusted the main models for regional background levels of $NO_2$ exposure and distance to major road in a sensitivity analysis. To check the effect modification by SES, we stratified the children residing in an urban area, based on parental education, into groups of low parental education (*n* = 82) and high parental education (*n* = 154). We investigated the association between green space and IQ in children of parents with a low education level and in children of parents with a high education level. As a final sensitivity analysis, we investigated the association between intelligence and residential green space based on the high-resolution model (2012).

## Results

### Characteristics of the study population

Table 1 summarizes the characteristics of the study population including 310 mothers and the 620 children. Based on the residential address of the twins at time of the IQ measurements,

**Table 1. Study population characteristics.**

| Characteristic | All *n* = 620 | Urban *n* = 236 | Suburban *n* = 128 | Rural *n* = 256 |
|---|---|---|---|---|
| Sex, male | 312 (50.3) | 114 (48.3) | 71 (55.5) | 127 (49.6) |
| Birth year | 1,986 ± 2.8 | 1,987 ± 2.7 | 1,987 ± 2.5 | 1,986 ± 2.9 |
| Zygosity-chorionicity | | | | |
| Dizygotic-dichorionic | 356 (57.4) | 150 (63.6) | 74 (57.8) | 132 (51.6) |
| Monozygotic-dichorionic | 106 (17.1) | 40 (16.9) | 14 (10.9) | 52 (20.3) |
| Monozygotic-monochorionic | 158 (25.5) | 46 (19.5) | 40 (31.3) | 72 (28.1) |
| Parental education | | | | |
| Low | 120 (38.7) | 41 (34.8) | 4 (6.3) | 59 (46.1) |
| High | 190 (61.3) | 77 (65.2) | 44 (68.8) | 69 (53.9) |
| Neighborhood income, euro | 18,831 ± 2,481 | 18,898 ± 2,426 | 19,755 ± 2,848 | 18,308 ± 2,196 |
| Intelligence (WISC-R) | *n* = 620 | *n* = 236 | *n* = 128 | *n* = 256 |
| Age at investigation | 11.0 ± 1.8 | 11.0 ± 1.7 | 10.9 ± 1.7 | 11.1 ± 1.9 |
| Year of investigation | 1,998 ± 1.6 | 1,998 ± 1.4 | 1,998 ± 1.6 | 1,998 ± 1.7 |
| Intelligence score (IQ) | | | | |
| Total | 105.3± 14.2 | 106.2 ± 13.2 | 104.8 ± 14.2 | 104.8 ± 15.1 |
| Verbal | 106.2 ± 13.5 | 107.0 ± 12.9 | 106.1± 13.3 | 105.5 ± 14.2 |
| Performance | 103.2 ± 15.2 | 103.9 ± 14.1 | 102.4 ± 15.7 | 103.0 ± 15.9 |
| Behavior (CBCL) | *n* = 442 | *n* = 168 | *n* = 92 | *n* = 182 |
| Age at investigation | 8.5 ± 2.2 | 8.4 ± 2.2 | 8.1 ± 2.0 | 8.7 ± 2.3 |
| Year of investigation | 1,994 ± 0.5 | 1,994 ± 0.5 | 1,994 ± 0.5 | 1,994 ± 0.5 |
| CBCL score | | | | |
| Total | 46.5 ± 9.9 | 47.0 ± 9.5 | 46.8 ± 9.7 | 45.9 ± 10.3 |
| Externalizing | 46.3 ± 9.9 | 47.2 ± 9.6 | 46.2 ± 10.1 | 45.4 ± 10.1 |
| Internalizing | 47.7 ± 9.4 | 48.0 ± 9.4 | 48.2 ± 9.4 | 47.0 ± 9.4 |

Data presented are means ± standard deviation or number (percentage).

CLBL, Child Behavior Checklist; WISC-R, Wechsler Intelligence Scale for Children-Revised.

236 (38%) children were living in an urban environment, 128 (21%) in a suburban area, and 256 (41%) in a rural area. Overall, most parents (61.3%) were highly educated. The population comprised 312 (50.3%) boys and 308 (49.7%) girls. Our analysis included 356 (57.4%) dizygotic twins, 106 (17.1%) monozygotic-dichorionic twins, and 158 (25.5%) monozygotic-monochorionic twins. The children completed the WISC-R test at a mean age of 11.0 years (range: 7–15 years). The averaged IQ scores (±SD) were 105.3 ± 14.2 for total IQ, 106.2 ± 13.5 for verbal IQ, and 103.2 ± 15.2 for the performance IQ. Behavior was assessed by the CBCL in a subset of 442 children. The mean CBCL total score was 46.5 ± 9.9, the mean CBCL externalizing score was 46.3 ± 9.9, and the mean CBCL internalizing score was 47.7 ± 9.4. Table 2 contains the residential distribution of green space for a 500–5,000-m radius. The median residential green space ranges from 1.13% to 3.78%. In the 500-m radius, green space was present for 17% of the participants (respectively, 19%, 20%, and 14% in urban, suburban, and rural) and in the 1,000-m radius for 37% of the participants (respectively, 45%, 42%, and 27% in urban, suburban, and rural).

## Intelligence in association with green space in childhood

We noticed (Fig 1) a significant urbanicity-by-green space interaction on twin IQ (*p* for interaction <0 .001). We therefore stratified the analysis by degree of urbanicity. Higher residential

**Table 2. Distribution of green space.**

| Green space | Urban | | | Suburban | | | Rural | | |
|---|---|---|---|---|---|---|---|---|---|
| | 25th percentile | Median | 75th percentile | 25th percentile | Median | 75th percentile | 25th percentile | Median | 75th percentile |
| 5,000-m buffer, % | 2.88 | 3.44 | 4.86 | 1.96 | 3.32 | 5.61 | 1.97 | 3.45 | 6.06 |
| 4,000-m buffer, % | 2.56 | 3.78 | 4.91 | 1.63 | 3.39 | 5.99 | 1.45 | 3.09 | 6.13 |
| 3,000-m buffer, % | 2.11 | 3.38 | 5.39 | 1.29 | 2.82 | 6.11 | 0.19 | 2.95 | 5.24 |
| 2,000-m buffer, % | 0.25 | 3.01 | 5.96 | 0 | 3.2 | 7.33 | 0 | 1.13 | 4.09 |
| 1,000-m buffer, % | 0 | 0 | 4.85 | 0 | 0 | 6.56 | 0 | 0 | 0.3 |
| 500-m buffer, % | 0 | 0 | 0 | 0 | 0 | 0 | 0 | 0 | 0 |

Green space was defined as seminatural, forested, blue, and urban green areas based on CORINE Land Cover 1990 (European Environment Agency).

CORINE, Coordination of Information of the Environment.

surrounding green space is associated with intelligence in children living in an urban environment before (Fig 1) and after adjustment (Fig 2) for sex, age, parental education, neighborhood household income, year of assessment of intelligence, and zygosity and chorionicity. This was not observed among children living in a rural or suburban environment. In urban areas, residential green space within a 1,000- to 5,000-m buffer is significantly associated with TIQ and PIQ, whereas the association between urban green space and verbal IQ was significant in a 2,000- to 5,000-m buffer and not in a 500-m and 1,000-m buffer. Taking the percentage green within a 3,000-m buffer as a reference, we found that for an IQR (3.3%) increase in green space within a 3,000-m radius from the residential address is associated with a 2.6-point higher (95% CI 1.4–3.9; $p < 0.001$) TIQ, a 2.2-point higher (95% CI 0.9–3.4; $p = 0.0008$) VIQ, and a 2.4-point higher (95% CI 1.0–3.8; $p = 0.0014$) PIQ among children residing in an urban environment. The full results of the model with all the covariates are shown in supplement (S1 Table).

Fig 3 shows the cumulative frequency distributions of total IQ scores in children residing in high green versus low green urban areas. Children in low green areas are more likely (4.2% versus 0%) to have a total IQ of 80 or lower. Furthermore, although 11.9% of those living in a green area had IQ scores in the superior range (>125), only 4.2% of the children living in a

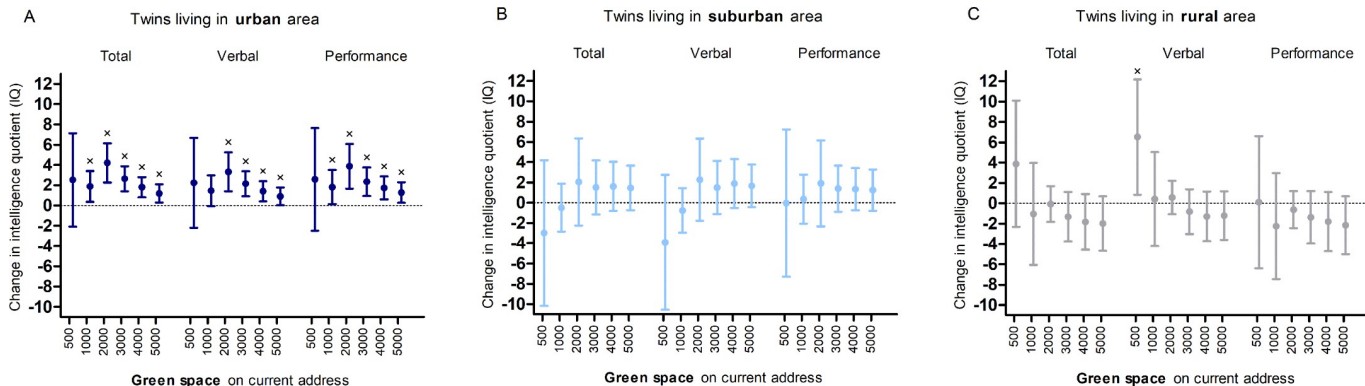

**Fig 2. Estimated change in IQ for an IQR increase in green space in a 500–5,000-m radius around the residence in twins living in (A) an urban ($n = 234$), (B) a suburban ($n = 128$), and (C) a rural area ($n = 252$) adjusted for sex, age, parental education, neighborhood household income, year of IQ test, and zygosity and chorionicity.** We expressed the estimates for an IQR change in percentage green space for buffers from 1,000 to 5,000 m. Because of the skewedness of the data for the 500-m radius (and 1,000 m in rural), we expressed the effect size for a dichotomized variable representing the presence or absence of green space within this buffer. IQ, intelligence quotient; IQR, interquartile range.

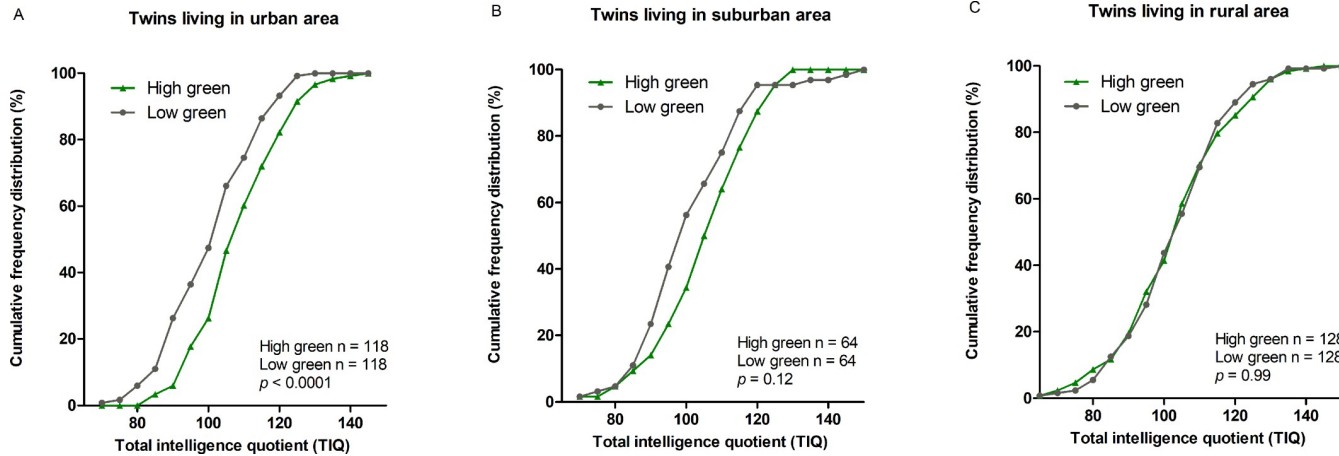

**Fig 3. This graph displays the cumulative frequency distribution of total IQ scores in children residing in high and low green space (based on the median) in twins living in (A) an urban, (B) a suburban, and (C) a rural area.** IQ, intelligence quotient; TIQ, total intelligence quotient.

low green area had scored in this range. We observed no difference in cumulative frequency distribution between high green and low green in children living in either rural or suburban areas.

Besides green space at the residential address in childhood, we also observed a significant association between green space during pregnancy with intelligence in children living in an urban area (S3 Fig).

To study the robustness of our findings, we performed multiple sensitivity analysis. The results remain similar in a subpopulation of children ($n = 144$) living a minimum of 10 years at the current address (S4 Fig). Additional adjustment for distance to major roads and regional background levels of $NO_2$ did not result in a considerable change in the aforementioned effect estimates (S5 Fig). We performed stratified analysis to study the potential effect modification by parental education. Patterns were similar according the parental educational level, but overall, we found a tendency of more pronounced associations between residential green space and IQ outcomes in urban children of parents with higher educational background (S6 Fig). Further, we investigated the association between intelligence and residential high green (vegetation height higher than 3 m) based on a high-resolution model. Our results (S7 Fig) show that high green is positively associated with VIQ and TIQ in large buffer sizes ($> 500$ m).

## Child behavior in association with green space

Similarly, higher residential green space is associated with a reduction in behavioral problems in children living in an urban area, whereas no effect was noted in children living in a rural or suburban area (Fig 4). Percentage of residential green space within a 1,000-m to 3,000-m radius from the residential address in an urban environment was significantly associated with a reduction in total problem behavior, green space within a 500-m to 4,000-m radius is significantly associated with a reduction in externalizing problem behavior (including attention problems and aggressive behavior) and green space in a 3,000-m radius is associated with lower internalizing problem behavior (including anxiety and withdrawal). An IQR (3.6%) increase in green space within a 3,000-m radius from the residential address is associated with a 2.0 (95% CI −3.6 to −0.4; $p = 0.015$) lower CBCL total score and a 2.0 (95% CI −3.5 to −0.4; $p = 0.017$) lower CBCL externalizing score among urban-residing children.

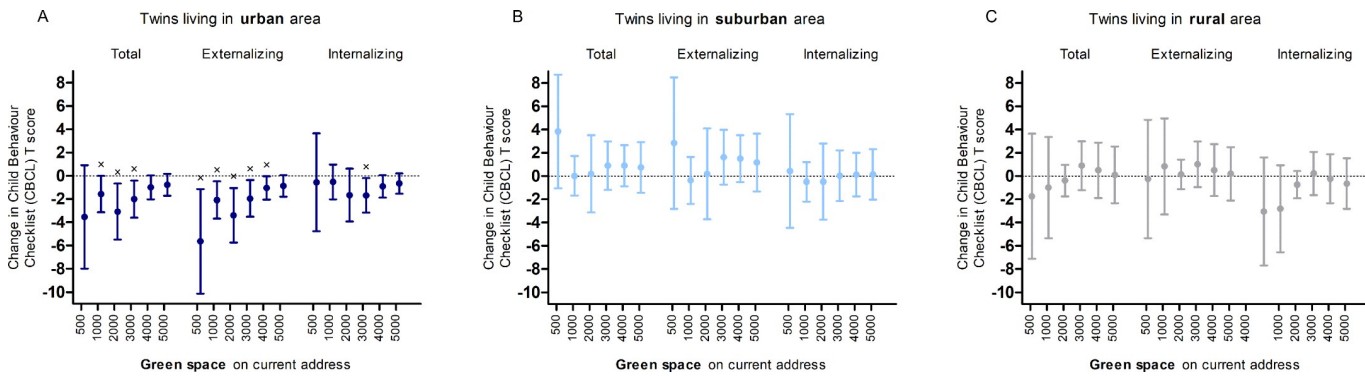

**Fig 4. Estimated change in CBCL score for an IQR increase in green space in a 500–5,000-m radius around the residence.** In twins living in (A) an urban (*n* = 168), (B) a suburban (*n* = 92), and (C) a rural area (*n* = 182). Adjusted for sex, age, parental education, neighborhood household income, month of CBCL, and zygosity and chorionicity. We expressed the estimates for an IQR change in percentage green space for buffers from 1,000 to 5,000 m. Because of the skewedness of the data for the 500-m radius (and 1,000 m in rural), we expressed the effect size for a dichotomized variable representing the presence or absence of green space within this buffer. CBCL, Child Behavior Checklist; IQR, interquartile range.

## Discussion

The key finding of our analysis is that a higher percentage of residential green space is associated with higher intelligence and lower behavioral problems in 7–15-year-old children living in urban areas. More particularly, the level of urbanicity modified the association of green space exposure with intelligence and behavior in childhood.

To our knowledge, this is the first study investigating the association between residential green space and intelligence in children. Previous studies have already shown that urban green is important for cognitive development in children by improving working memory, attention, and school performance. A study in schoolchildren in Barcelona [29] shows that school and total surrounding green space were associated with enhanced progress in working memory and superior working memory and a greater reduction in inattentiveness. The same investigators found in a subcohort of 253 children, using high-resolution 3-D MRIs, several brain regions that had larger volumes in association with higher lifelong exposure to residential surrounding greenness [56]. In a prospective birth cohort study, Dadvand and colleagues established that higher lifelong residential surrounding green space was associated with better scores on tests of attention at 4–5 years and 7 years of age [57]. In addition, based on school-level student performance data from 905 public schools in Massachusetts (the second most tree-covered state) collected between 2006 and 2012 (*n* = 6,333), the effect of surrounding greenness on the school performance was investigated [34]. The results showed that children (age 8–9 years) with higher exposure to greenness show better performance in both English and math. However, this study was replicated in Chicago, in an area with low tree coverage, and found no convincing evidence for a positive relationship between residential greenness and academic performance [58]. The authors suggest that a main reason may be that greenness, the general degree of vegetation, based on NDVI (normalized difference vegetation index) fails to distinguish between vegetation types, such as grasses and trees. In connection to this, Kweon and colleagues found a positive association between tree cover in the school surrounding and academic performance, but grass and shrubs negatively correlate with performance [59].

Few studies on green space and academic performance have specifically tested for effect modification by urbanicity. A cross-sectional study in Maryland public schools tested the relationship between 5 greenspace measures and performance (*n* = 668) around schools and in children's neighborhoods [39]. Including interaction terms in the models showed no clear

effect modification by urbanicity. We show that an IQR (3.3%) increase in green space within a 3,000-m radius from the residential address is associated with a 2.6-point higher (95% CI 1.4–3.9; $p < 0.001$) total IQ in childhood. The magnitude of our estimates show that our findings are relevant for public health. Indeed, although the contribution on the average population IQ might be moderate, children with low residential green space are more likely to gravitate to the lower tail of the IQ curve. Therefore, low residential green space in children living in an urban area might cause an inevitable "shift" in population IQ. This has been demonstrated by Needleman for lead exposure during childhood, showing that a greater proportion of children with high lead exposure gravitate to the lower tail of the normal IQ curve [60], and here, we show the same phenomena for low residential green space exposure. Indeed, 4.2% of the children had an IQ below 80 growing up in low urban green space areas versus 0% in high green space. This shift towards a higher incidence of a low IQ demonstrates the public health impact of our findings at an unprecedented level.

Besides green space at the residential address in childhood, we also observed a significant association between green space during pregnancy with intelligence in children living in an urban area. Until now, only 1 study in 1,312 women and their children investigated surrounding green spaces at the residential address at birth. It shows that exposure to higher levels of surrounding green spaces is associated with better early childhood cognition at the age of 2 [61].

Our findings show that residential green space is associated with a reduction in total and mostly externalizing problem behavior (including attention problems and aggressive behavior) in 7–15-year -old children living in a urban area. This in line with other studies on green space using the CBCL to assess behavior. School children 6–18 years of age living in the highest tertile of greenness had a lower total CBCL score, especially aggressive behavior and attention problems, compared with children living in the lowest tertile of greenness [40]. As in our study, the researchers found a slightly stronger association between greenness and externalizing behavior than with internalizing behavior [40]. In urban-dwelling adolescents, exposure to neighborhood greenspace is associated with reducing aggressive behavior as reported by parents on the aggression subscale of the CBCL [38]. A systematic review present evidence for an association between green space exposure and the total difficulties score of the Strengths and Difficulties Questionnaire (SDQ), a measure of general mental health in children [62]. Regarding the different subdomains of the SDQ, 5 out of 6 studies documented significant associations with hyperactivity and inattention problems in association with green space exposure [36,37,63–65]. A study in 4–13-year-old children shows that green space quality and quantity were associated with internalizing and externalizing subscale scores for all age groups [66]. The participants in the study of Markevych and colleagues (2014) were spatially spread over the inner city of Munich and its surrounding areas [37]. The analyses stratified by whether or not a child lived in the inner city, showed that the associations between urban green spaces and lower behavioral problems was slightly stronger for children living in the inner city ($n$ = 782) compared with those living in the surrounding suburban areas ($n$ = 1,150); however, the interaction-term was not significant.

Several reasons can be postulated why no association is found between green space and IQ as well as behavioral problem scores in suburban and rural residents compared with urban residents. First, Markevych and colleagues (2014) suggest that there is a general lack of green spaces in urban areas, and thus, parks play a more important role for urban residents than for those living in suburban or rural areas, in which natural green spaces are more prevalent [37]. A previous study [67] shows that the association between green space and health varied according to the combination of urbanicity and area income deprivation. A possible explanation is that high-income residents in suburban and rural areas have their own domestic gardens, and municipal green space is thus less important to them [67]. Finally, rural areas might

have fewer everyday destinations (e.g., grocery store, post office, and pharmacy) within walking distance, resulting in more car dependency and less active transportation [68]. Therefore, urban residents may have more benefits of green space in their neighborhood.

The underlying pathways responsible for the positive effect of residential green space exposure on intelligence is not fully understood, but several mechanisms are considered. First, green spaces provide environmental benefits (e.g., reducing exposure to air pollution, noise, and heat). Moreover, green spaces encourage health-promoting activities and facilitate social cohesion. A third potential domain linking green space to health is the ability of green space to restore capacities (e.g., attention restoration and physiological stress recovery) [68].

Our results were observed in twins. We do not expect that twins have a different exposure or react differently to green space than singletons. Studies show that twins score 4 points lower on an IQ test than singletons [69]. This difference is partially attributed to difference in intra-uterine growth [69], nutrient deficiency, and shorter gestational length of twins [70]. In another study, however, no evidence on cognitive differences was found between adult twins and their nontwin siblings [71]. Besides, lower birth weight is a causal risk factor for child problem behavior [72]. However, a study comparing the developmental trajectories of behavioral problems in 6- to 12-year-old twins and singletons confirmed the generalizability with regard to externalizing problems in childhood of twin studies to singleton populations [73]. Therefore, we expect no large differences in regard to singletons.

We recognize limitations of the current study. First, surrounding green space was assessed based on residential location; however, no information on school location was available. However, children in Flanders (Belgium) usually attend schools near their home, on average, 2.8 km. Therefore, a buffer distance such as 3,000 m and more may reflect the full exposure to green space that is part of the residence and the school neighborhood where children likely spend their time [34]. Second, the twins were born from 1980 to 1991; however, to determine residential green space during pregnancy, we used data from 1990 as no earlier satellite data were available. However, strong correlations have been shown over time between periods of 1 decade [22]. The association between green space and the outcomes IQ and behavior was investigated in several radius distances around the residential address. We found that the positive impact of green in an urban setting for both IQ and behavioral scores depended on the percentage green within 2,000 and 3,000 m. It was unfeasible to choose buffers smaller than 500 m given the resolution of CORINE Land Cover data. However, our results were comparable when we use high-resolution maps (S7 Fig). Third, no information was available on time–activity patterns, such as time spend outdoors, and on possible mediators between green space and intelligence. An important strength of our study is that we excluded potential confounding by socioeconomic indicators and traffic-related air pollution. Indeed, we presented our associations independently of socioeconomic indicators both at the individual level, such as parental education, as well as by group level by including neighborhood household income. Furthermore, in a sensitivity analysis, we account for residential air pollution and residential distance to major road as a proxy for traffic-related noise and air pollution. The proxy of residential distance to a major road is currently still used in environmental epidemiology [74–76] and is correlated with black carbon load in children's urine [77].

The results of this study provide important policy and public health implications. Whereas in 1950, only 30% of the world's population lived in urban areas; nowadays, this is already more than half of the global population, and it is expected to increase to 68% by 2050 [78]. Urban areas are characterized by a network of nonnatural built-up infrastructures where residents often have limited access to natural environments in their daily lives [56,79]. Understanding the health disparities that exist between urban and rural environments is essential for maintaining and improving human well-being in a rapidly urbanizing world [33].

## Conclusions

Our results indicate that residential green space may be beneficial for intellectual and behavioral development of children living in an urban environment. An important contribution of this paper is that to our knowledge this is the first study on intelligence and residential green space. We showed a shift in the IQ distribution of urban children in association with residential green space exposure within 3,000-m radius. These findings are relevant for policy makers and urban planners designing urban environments resulting in a sustaining effective policy. Well-planned cities can offer unique opportunities to create an optimal environment for children to develop their full potential.

## Supporting information

**S1 STROBE Checklist. The checklist of STROBE guidelines.** STROBE, Strengthening the Reporting of Observational Studies in Epidemiology.
(DOCX)

**S1 Text. Questionnaire.**
(PDF)

**S2 Text. A priori statistical analysis plan.**
(PDF)

**S1 Table. Full model with all covariates of the association between green space in 3,000-m radius around the residence and intelligence.** CI, confidence interval; IQR, interquartile range; TIQ, total intelligence quotient; TPIQ, performance intelligence quotient; TVIQ, verbal intelligence quotient. *low versus high.
(DOCX)

**S1 Fig. Statistical sectors in East Flanders classified as urban, suburban and rural areas depending on their population density, employment, location, and spatial planning.** Source: Flemish Government–Department Environment.
(TIF)

**S2 Fig. Flowchart.** From the children with IQ data available, 599 mothers were invited to complete a questionnaire on residential history. A total of 324 mothers completed the questionnaire. From the twin pairs, we excluded 14 pairs because of missing data on residential location or parental education resulting in a final IQ study sample of 620 children. For a subset of 442 children, aged 6–15 years, data are available for studying behavioral characteristics. IQ, intelligence quotient.
(TIF)

**S3 Fig. Estimated change in IQ for an IQR increase in green space in a 1,000–5,000-m radius around the residence during pregnancy and in childhood.** Dark blue: Residential green space in childhood at time of IQ measurement ($n$ = 236) adjusted for sex, age, parental education, neighborhood household income, year of IQ test (linear and quadratic), and zygosity and chorionicity. Red: Residential green space during pregnancy ($n$ = 216) adjusted for sex, gestational age, birth weight, birth year (linear and quadratic), zygosity and chorionicity, cord insertion, parental education, neighborhood household income, and smoking during pregnancy. IQ, intelligence quotient; IQR, interquartile range.
(TIF)

**S4 Fig. Estimated change in IQ for an IQR increase in green space in a 1,000–5,000-m radius around the residence in children living in an urban ($n$ = 236) area and in a subgroup**

**of children living minimum 10 years (or their whole life if the age of the child <10 years old) at their current address (*n* = 144).** Adjusted for sex, age, parental education, neighborhood household income, year of IQ test, and zygosity and chorionicity. IQ, intelligence quotient; IQR, interquartile range.
(TIF)

**S5 Fig. Main model additionally adjusted for distance to major road and NO2 exposure.** Estimated change in IQ for an IQR increase in green space in a 1,000–5,000-m radius around the residence in children living in an urban (*n* = 236) area. The main model is adjusted for sex, age, parental education, neighborhood household income, year of IQ test, and zygosity and chorionicity. We additionally adjust for distance to major road and regional background levels of NO2 exposure the year before IQ test. IQ, intelligence quotient. IQ, intelligence quotient; IQR, interquartile range; NO2, nitrogen dioxide.
(TIF)

**S6 Fig. Effect modification socioeconomic status.** Estimated change in IQ for an IQR increase in green space in a 1,000–5,000-m radius around the residence in children living in an urban area. Based on parental education, we stratified the children residing in an urban area into groups low parental education (*n* = 82) and high parental education (*n* = 154). Adjusted for sex, age, parental education, neighborhood household income, year of IQ test, and zygosity and chorionicity. IQ, intelligence quotient; IQR, interquartile range.
(TIF)

**S7 Fig. Estimated change in IQ for an IQR increase in green space (high-resolution model 2012) in a 50–2,000-m radius around the residence.** Green space is high green (vegetation height higher than 3m, including blue space). In twins living in (A) an urban (*n* = 236), (B) a suburban (*n* = 128), and (C) a rural area (*n* = 256). Adjusted for sex, age, parental education, neighborhood household income, year of IQ test, and zygosity and chorionicity. IQ, intelligence quotient; IQR, interquartile range.
(TIF)

## Author Contributions

**Conceptualization:** Esmée M. Bijnens, Catherine Derom, Evert Thiery, Steven Weyers, Tim S. Nawrot.

**Data curation:** Esmée M. Bijnens, Catherine Derom.

**Formal analysis:** Esmée M. Bijnens.

**Funding acquisition:** Esmée M. Bijnens, Catherine Derom, Evert Thiery, Tim S. Nawrot.

**Investigation:** Esmée M. Bijnens.

**Supervision:** Catherine Derom, Evert Thiery, Steven Weyers, Tim S. Nawrot.

**Visualization:** Esmée M. Bijnens.

**Writing – original draft:** Esmée M. Bijnens, Tim S. Nawrot.

**Writing – review & editing:** Catherine Derom, Evert Thiery, Steven Weyers.

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
