## [Decision Letter · Decision Letter 0]

13 Mar 2020

Dear Dr. Bijnens,

Thank you very much for submitting your manuscript "The importance of residential greenness on intellectual and behavioural development in children living in urban versus rural environments" (PMEDICINE-D-19-03156) for consideration at PLOS Medicine. Please accept my sincere apologies for the unusual delay in getting back to you about it. 

[LINK]

In light of these reviews, I am afraid that we will not be able to accept the manuscript for publication in the journal in its current form, but we would like to consider a revised version that addresses the reviewers' and editors' comments. Obviously we cannot make any decision about publication until we have seen the revised manuscript and your response, and we plan to seek re-review by one or more of the reviewers. 

We expect to receive your revised manuscript by Apr 03 2020 11:59PM. Please email us (plosmedicine@plos.org) if you have any questions or concerns.

We look forward to receiving your revised manuscript. 

Sincerely,

Clare Stone, PhD

Managing Editor 

PLOS Medicine

plosmedicine.org

Please revise your title according to PLOS Medicine's style. Your title must be nondeclarative and not a question. It should begin with main concept if possible. "Effect of" should be used only if causality can be inferred, i.e., for an RCT. Please place the study design ("A randomized controlled trial," "A retrospective study," "A modelling study," etc.) in the subtitle (ie, after a colon).In addition, as the clarity around the use of ‘greenness’ has been raised by referees, please also consider this and please add the country setting. 

Abstract – Please start the ‘Methods and Findings’ section with more general background about the study rather than the intelligence scale used (focus should be more on how selection took place, numbers, dates, locations and some brief demographic information); please add a sentence on the limitations of the study as the final sentence or two of the Methods and Findings section. 

Data - PLOS Medicine requires that the de-identified data underlying the specific results in a published article be made available, without restrictions on access, in a public repository or as Supporting Information at the time of article publication, provided it is legal and ethical to do so. Note Corresponding authors cannot be points of contact for data requests. Please see the policy at 

http://journals.plos.org/plosmedicine/s/data-availability

and FAQs at 

http://journals.plos.org/plosmedicine/s/data-availability#loc-faqs-for-data-policy

Line 109 – please ensure all questionnaires are provided (and translated to English, as needed) as Supp Files and provide a callout in the methods. 

Line 113 – what is the ‘Child Behaviour Checklist’

Line 228 – remove ‘till now’ and instead replace with ‘to our knowledge this is the first…’

Be cautious around overstating findings (eg in conclusion…’ is especially’ ….may be would be better as this isn’t a trial)

Did your study have a prospective protocol or analysis plan? Please state this (either way) early in the Methods section.

Please ensure that the study is reported according to the STROBE guideline, and include the completed STROBE checklist as Supporting Information. Please add the following statement, or similar, to the Methods: "This study is reported as per the Strengthening the Reporting of Observational Studies in Epidemiology (STROBE) guideline (S1 Checklist)."

Comments from the reviewers:

Reviewer #1: See attachment

Michael Dewey

Reviewer #2: This study is among the first studies on IQ and greenspace, and is among the few existing studies on behavioural problems and greenspace. After revision, it may be considered for publication. My specific comments and recommendations are outlined below.

L67-69: The Authors never explain what is greenness, and instead, go to green spaces right away. These are two different concepts, which shuld be at least briefly mentioned. Greenness referes to general degree of vegetation while by green spaces, structured vegetation in parks, forests, meadows, etc. is meant. Greenness is typically satellite-derived and green spaces are extracted from land use/land cover datasets (if these data are not self-reported, of course).

L73-80: Talking about greenspace and academic performance, Authors ignore negative findings, as summarized by https://www.ncbi.nlm.nih.gov/pmc/articles/PMC6388261/. Moreover, the analysis of Wu et al. had methodological shortcomings, judging by the re-analysis of the data (https://www.sciencedirect.com/science/article/pii/S0169204618303086).

L69-71: Pathways explaining effects of green spaces/greenness are outlined here https://pubmed.ncbi.nlm.nih.gov/28672128-exploring-pathways-linking-greenspace-to-health-theoretical-and-methodological-guidance/ and here https://pubmed.ncbi.nlm.nih.gov/24387090-nature-and-health/.

L106: Please correct the reference. It is Supplement Figure 1, not Figure 1. 

L104: Just to make it clear: children born from 1964 were included into the analysis? It does not seem so from the further text.

L121: A very brief explanation that monochorionic twins share the same placenta would be nice for readers without medical background.

L133-134: Was complete residential history unknown? At least, time lived at the current address? How about moving during pregnancy? All this may have caused exposure misclassification.

L138-141: 

(1) How did this correspond to the years of birth and IQ assessment? Please provide the information on when the children were born and tested. Land cover did not remain absolutely the same over decades. 

(2) Furthermore, why not smaller buffers were considered? At least 500-m buffer would be informative. If distribution is too skewed (seems so), I would recommend to dichotomize the variabe into presence vs absence of green spaces in the given buffer. It is very hard for me to believe that green space in such big buffers would be more important for children than green space close to their home/school.

(3) Also, CORINE's minimum mapping unit is 25 ha which means that smaller parks are not present in those dataset. Some Authors even say that CORINE is not exactly applicable if urban areas (e.g., https://bmcpublichealth.biomedcentral.com/articles/10.1186/1471-2458-12-337), which I agree with. Have Authors tried using Landsat-derived vegetation indexes (NDVI, SAVI, EVI, etc.) as a sesnsitivity analysis? I would be curious to see whther results are similar. 

(4) Finally, since Authors dealt with CORINE-derived green spaces, I would rename the article and use „green spaces" instead of „greenness" throughout the manuscript.

L144-159: What are minimum and maximum scores of the both WISC-R and CBCL tests, in total and od the subscales? This information would be of interest for a reader.

L171, L173: Please briefly mention why quadratic year was added to the models. 

Have the Authors checked non-linearity of the studied associations somehow? This might be a reason for not seeing some associations. At least, I would recommend to categorize green space variables in tertiles in a sensitivity analysis.

L172: Why models were adjusted for distance to major road (and how this major road was defined?)? As a proxy for air pollution and/or noise? In this case, I would suggest to at least look at the effect estimates without this variables (if the Authors believe air pollution and noise are confounders) or even to do mediation analysis (if the Authors think this is mediation).

Table 1: How exactly parental education was defined - as the highest education of both parents baed on the available information? Please clarify in the text.

L203 and Figure 2: The strongest association was actually with 2000m buffer.

To all figures: Please consider using different symbols rather than colours, as many people have problems with differentiating colours. Figure 4 caused me eye pain, for instance.

L228-233: Actually, there is no associations overall, only in urban children. The three sentences should be re-formulated accordingly.

L237: This paper can also be cited: https://ehp.niehs.nih.gov/doi/full/10.1289/EHP1876?url_ver=Z39.88-2003&rfr_id=ori:rid:crossref.org&rfr_dat=cr_pub%20%200pubmed

L237-243: This is pretty much a repetition of L71-77.

L261: This study should also be cited: https://www.ajpmonline.org/action/showPdf?pii=S0749-3797%2817%2930377-X

L266-267: Markevych et al. (ref. 17 in your list) also had urban and suburban-rural composition of the study area, speaking of urbanicity. There are many more studies on other outcomes that investigated effect modification by urbanicity, just from what I can easily remember: https://www.sciencedirect.com/science/article/pii/S1353829218311158 and https://www.mdpi.com/1660-4601/13/3/311/htm.

L270-272: In this context, I would especially encourage the Authors to consider some satellite-derived vegetation index, which would capture all vegetation, not only green spaces min 25 ha in area.

L273-300: This section is too long considering that the Authors did not attempt to investigate any of the mechanisms by mediation analyses. I would suggest condensing this part.

L305-307: This is not true. Landsat 4-5 TM satellite images are available since early 1980s, and they could have been used in this study in addition to CORINE 1990.

Among the limitations, which are clearly more than two: poor quality land cover data to derive green spaces data, as specified before, unavailability of full residential history and time-activity pattern, absence of mediation analysis. Even effect modification by SES was not checked.

L309: Additional adjustment for distance to major road does not exclude confounding by air pollution, as distance to major road is just a mere proxy. I wonder if real air pollution data were not available. For example, there are ozone and NO2 maps from the ELAPSE project for the year 2000, and the estimates can be back-extrapolated. In any case, I would at least soften the language in this part. 

More effort should be put into explaining why green space was not important for suburban and rural residents compared to urban residents. Also, why no associations were observed for internalizing problems? Why not for all buffers associations were present? In my view, Discussion needs to be shorten in some parts and elaborated in others.

[LINK]

---

## [Decision Letter · Decision Letter 1]

3 Jun 2020

Dear Dr. Bijnens,

Thank you very much for re-submitting your manuscript "The importance of residential green space on intellectual and behavioural development in children living in urban versus rural environments in Belgium; a longitudinal birth cohort study" (PMEDICINE-D-19-03156R1) for review by PLOS Medicine.

I have discussed the paper with my colleagues and the academic editor and it was also seen again by the previous reviewers. I am pleased to say that provided the remaining editorial and production issues are dealt with we are planning to accept the paper for publication in the journal.

[LINK]

We look forward to receiving the revised manuscript by Jun 10 2020 11:59PM. 

Sincerely,

Clare Stone, PhD

Managing Editor 

PLOS Medicine

plosmedicine.org

Requests from Editors:

We would ask that the title is revised to: "Residential green space and child intelligence and behaviour across urban, suburban and rural areas in Belgium: a longitudinal birth cohort study of twins"

Please list the adjustment factors in the abstract

Around line 50, please quote at least one further limitation, e.g. unmeasured confounding (it seems likely that confounding would be an issue in this study)

Please make changes to statistical values: p<0.0001->p<0.001 in the results, and elsewhere

Please ensure the STROBE checklist is in sections and paragraphs instead of page numbers as these can change during formatting etc. 

Please confirm that the email address for data (twins@uzgent.be.) does not direct to one of the authors on this manuscript, as this is not permitted. If so, please find another point of contact or URL as for the other data points in the study.

Please remove ‘Statements’ (lines 450-474) from the Word doc as these will be pulled in automatically from information in EM.

Comments from Reviewers:

Reviewer #1: The authors have addressed all my points thoroughly.

Michael Dewey

Reviewer #3: 

after reviewing all the answers of the comments, I consider authors made all the modifications required and therefore, I would agree continuing with the proccess.

[LINK]

---

## [Editor Report · Decision Letter 2]

8 Jul 2020

Dear Dr. Bijnens, 

On behalf of my colleagues and the academic editor, Dr. Iana Markevych, I am delighted to inform you that your manuscript entitled "Residential green space and child intelligence and behaviour across urban, suburban and rural areas in Belgium; a longitudinal birth cohort study of twins" (PMEDICINE-D-19-03156R2) has been accepted for publication in PLOS Medicine. 

PRODUCTION PROCESS

PRESS

PROFILE INFORMATION

Thank you again for submitting the manuscript to PLOS Medicine. We look forward to publishing it. 

Best wishes, 

Clare Stone, PhD

Managing Editor 

PLOS Medicine

plosmedicine.org